# Manufacturability and Stress Issues in 3D Silicon Detector Technology at IMB-CNM

**DOI:** 10.3390/mi11121126

**Published:** 2020-12-18

**Authors:** David Quirion, Maria Manna, Salvador Hidalgo, Giulio Pellegrini

**Affiliations:** Instituto de Microelectrónica de Barcelona, IMB-CNM (CSIC), Campus UAB, 08193 Barcelona, Spain; maria.manna@imb-cnm.csic.es (M.M.); salvador.hidalgo@imb-cnm.csic.es (S.H.); giulio.pellegrini@imb-cnm.csic.es (G.P.)

**Keywords:** 3D silicon detectors, process integration, process-induced stress, silicon manufacturing, radiation-hard detectors, high-energy physic experiments, through-silicon vias

## Abstract

This paper provides an overview of 3D detectors fabrication technology developed in the clean room of the Microelectronics Institute of Barcelona (IMB-CNM). Emphasis is put on manufacturability, especially on stress and bow issues. Some of the technological solutions proposed at IMB-CNM to improve manufacturability are presented. Results and solutions from other research institutes are also mentioned. Analogy with through-silicon-via technology is drawn. This article aims at giving hints of the technology improvements implemented to upgrade from a R&D process to a mature technology.

## 1. Introduction

3D silicon detectors [1,2] established themselves as a key technology in high-energy physics applications. Contrary to the planar detectors, the electrodes (junctions and Ohmic contacts) in a 3D detector are implemented in the bulk, and not the surface, of silicon (see Figure 1). This distinctive geometry allows these detectors to combine high radiation hardness and low power consumption. The high radiation hardness stems from the short distance between electrodes, limiting therefore the carrier trapping by radiation-induced defects, while the signal is defined by the larger track of the particles through the silicon wafer thickness. The low depletion voltage guarantees a low power consumption even at the over-depletion voltage needed to improve the charge collection. Both aspects are crucial in the inner tracking detectors of high-energy physics experiments. 3D silicon detectors are therefore used as pixel detectors at the closest to the interaction point or to the beam. They were successfully installed in 2014 in the Insertable B-Layer of the ATLAS detector (IBL-ATLAS) [3], in 2016 in the ATLAS Forward Proton (AFP) detector [4], in 2017 in the CMS-TOTEM Precision Proton Spectrometer (CT-PPS) [5] and are scheduled for the High Luminosity LHC (HL-LHC) upgrade, both in ATLAS and CMS detectors [6,7]. Their combined radiation hardness and timing properties make them also a promising alternative to planar detectors, which are limited to fluence of ~10^15^ cm^−2^ [8], and could establish themselves as reference solid-state timing detectors in harsh environment [9].

These appealing features, however, come at the expense of non-uniform signal, large sensor capacitance, due the small inter-electrode spacing and long electrode depth, and increased complexity of fabrication. Actually, the IBL technical design report stated that the *“main concerns specific to 3D sensors are manufacturability and uniformity of a production run*” [10].

3D technology is a relatively complex technology: a fabrication run consists of ~120–140 steps with 8 mask levels, compared to ~40 steps and 5 mask levels for standard planar pixels (according to IMB-CNM definitions). Manufacturability of 3D detectors was demonstrated during the IBL production ([3]; see also chapter 7 in [2]). This article proposes to give hints in the long way from a R&D process to a mature technology, and, more specifically, will address the influence of wafer bow on manufacturability. In the last ten years, on-line bow monitoring was systematically integrated to the standard process control at IMB-CNM. It allowed us to point out critical processes in terms of stress and to improve greatly the manufacturability of 3D detectors.

This article starts with an introduction to stress and wafer bow in silicon manufacturing (Section 1). Then, we discuss the general consequences of wafer curvature on manufacturing (Section 2). Finally, an overview of fabrication technologies for 3D detectors is given. For each technology, on-line bow monitoring is presented, as well as process improvements we developed to reduce stress (Section 3).

## 2. Stress and Wafer Bow in Silicon Manufacturing

### 2.1. Basics of Stress in Semiconductor Manufacturing

Almost any films deposited or grown on a wafer will be in a state of stress, be it because of the difference between deposition and room temperature (thermal stress), or from growth or deposition conditions (intrinsic stress). As stated by Doerner and Nix [11], “*stresses in thin films on substrates are produced by processes which would cause the dimensions of the film to change if it were not attached to the substrate.*” Let us imagine that the thin film dimensions would be larger than the substrate if not attached to it (Figure 2a). Attached to the substrate, the thin film would tend to expand (buckling up, compressive state), creating a stress of opposite sign at the substrate surface, and the substrate will bend upwards in order to balance the bending moment which is produced by the stressed film [12]. In the opposite case, the thin film will tend to crack (tensile state) and the substrate will bend downwards (Figure 2b).

Curvature *K* is the inverse of the curvature radius *R* of the wafer: *K* = 1/*R*. Townsend et al. [13] demonstrated that the total curvature *K* of a multilayer on a thick substrate is the sum of the individual curvatures *K_i_* induced by each layer composing the multilayer. Stresses in the substrate *σ_s_* and in each film *σ_i_* can be expressed in function of these curvatures respectively:(1)$$\sigma_{s}=\frac{E_{s}}{1-\nu_{s}}\left(\frac{t_{s}}{3}-z\right)K$$
(2)$$\sigma_{i}=\frac{1}{6}\frac{E_{i}}{1-\nu_{i}}\frac{t_{s}^{2}}{t_{i}}\left(K_{i}-K_{i-1}\right)$$

Subscripts *i* and *s* refer respectively to layer *i* and substrate. *E* represent the Young modulus and *ν* the Poisson ratio, *E*/(1 − *ν*) is the biaxial elastic modulus (180 GPa for Si (100) [14]), *t_i_* and *t_s_* are the layer and substrate thickness, respectively. *z* is the coordinate distance normal to the linear dimension of the composite and calculated from the backside of the substrate (the multilayer being deposited on the front side of the substrate). *K*_*i*-1_ and *K_i_* are the curvature of the wafer before and after deposition of layer *i* respectively. These results are valid within the thin film approximation: substrate thickness is much larger than film thickness, *t_i_ << t_s_*. Equation (2) is the Stoney equation [15].

The stress formula can be simplified to give a direct relation between stress and sagitta, or bow. Bow x is defined as the distance between the point at mid-thickness and wafer centre, and a reference plane defined by three points at the wafer edges, with the wafer in the free (not clamped) state (see Figure 3). The stress-thickness product of the *i*th layer is proportional to the bow difference *x_i_*-*x_i_*_−1_ before and after deposition of layer *i*:(3)$$\sigma_{i}.t_{i}=-\beta_{S}.\left(x_{i}-x_{i-1}\right)$$
with
(4)$$\beta_{S}=\frac{4}{3}\frac{E_{S}}{1-\nu_{S}}{\left(\frac{t_{S}}{y_{S}}\right)}^{2}$$

*β**_s_* is a parameter depending only on the substrate and so-to-say represents its rigidity. The minus sign comes from the definition of bow, as positive for compressive stress and negative for tensile stress. In the rest of this paper, we will only plot bow instead of product σ.t or the stress in each layer because we found it more practical and visual. It corresponds to a real distance, which affects directly the wafer manufacturability.

It is clear from Equation (3) that zero bow does not mean null stress in each layer, but a compensation of stress in the multilayer system.

Note that an identical layer (same stress-thickness curve) would induce a bow double on a 200 μm than on a 285-μm thick wafer, and five times from 230 μm to 525 μm (standard thickness for 100 mm wafers–see Figure 4).

The origin of stress is two-fold: intrinsic or growth residual stress, and thermal stress. The latter can be written:(5)$$\sigma_{i}=\frac{E_{i}}{1-\nu_{i}}\left(\alpha_{s}-\alpha_{i}\right)\left(T_{proc}-T_{meas}\right)$$

The subscript s and *i* refers as before to the substrate and the layer. α is the coefficient of linear thermal expansion (CTE). Values of CTE and biaxial elastic module B = E/(1 − ν) for typical materials in detector technology are given in the Table 1 below.

### 2.2. Bow Measurement at Microelectronics Institute of Barcelona (IMB-CNM)

Wafer bow was measured at Microelectronics Institute of Barcelona (IMB-CNM) with the Proforma300 from MTI Instruments, subsidiary of Mechanical Technology Incorporated [16]. It is a simple, manual, capacitance-based, non-contact system which allows to measure wafer thickness, total thickness variation (TTV), and bow. It measures the capacitance between the gauge and the wafer, and therefore calculates the distance between both (sensing gap), and then the wafer thickness. Wafer bow is measured with a 3-point fixture, which defines the reference plane. The wafer is first measured backside up, then turned over, and measured again. From the two measured distances, bow is easily deduced (see Figure 5 below). This procedure gives the bow with the sign convention defined above.

This system is appropriate for in-line measurement during wafer fabrication: it is non-contact and implies a minimum manipulation of the wafers.

Caveat: The measurement performed is a one-point measurement: only one distance at the wafer center is measured, and from it, bow is deduced. We did not measure the wafer general shape. Warping, which includes concave and convex regions, is therefore excluded from the principle. We performed various crosschecks with independent measurements of wafer shape (profile-meter, confocal microscopy) and constantly observed bow, and no warping.

## 3. Consequences of Bow on Manufacturability

For many years in microelectronic industry, wafer thickness was chosen and driven by experience of each foundry for specific technologies [17]. In high-energy physics applications, wafer thickness for detectors is driven by physics considerations and imposed externally. The technologist must adapt the technology to the wafer thickness requirement.

What are the consequences of wafer curvature on the manufacturability of the wafers? We will only list two of them here, which are of most importance for our technologies: alignment issues in photolithography and wafer breakage. We will conclude this part with a toy model to describe bow variation during the fabrication of a planar pixel detector.

### 3.1. Photolithography

One of the major threats of bow coming into mind is its effect on photolithography [18]. During mask exposition, wafers are clamped with a vacuum chuck. Thus the question is not the effect of bow on photolithography, but how out-of-plane distortion (OPD, or bow) translates into in-plane-distortion (IPD–see Figure 6 below). Recently, several groups tackled this issue in the more exigent context of advanced lithography nodes [19,20], but their conclusions can be used to our technology. Turner et al. [19] calculated from a simple analytic model that, in the case of uniform stress, the in-plane-distortion at step N + 1 ΔL_S(N+1)_ of two patterns separated by L is proportional to the wafer thickness t_s_ and the difference in curvature between step N + 1, K_N+1_, and step N, K_N_:(6)$$\Delta L_{S\left(N+1\right)}=-\frac{t_{S}L}{6}\left(K_{N+1}-K_{N}\right)$$

Considering the same assumptions as before, it is possible to calculate the in-plane-distortion as a function of the wafer bow Δ*x*:(7)$$\Delta L_{S\left(N+1\right)}=-\frac{4}{3}\frac{t_{S}L}{y^{2}}\Delta x$$

Assuming a standard 1 μm overlay error in our mask aligners, and that errors add quadratically, one can estimate the total overlay error in function of wafer bow between two photolithography levels, as plotted in Figure 6 below.

It is observed that the thicker the wafer, the larger the in-plane-distortion. This is due to the fact that the thicker the wafer, the larger the distance between the front surface, where the patterns lay, and the neutral plane, where stress and in-plane-distortion are null, which is located at 2/3 of the wafer thickness for a uniform stress [20].

Assuming that a 20% increase in overlay error is acceptable, it is seen that even in the worst case of a 450 μm thick wafer, a bow of 200 μm is still acceptable. Another practical limitation plays a role: we noticed that 100 mm wafers with bow superior to 300 μm could not be clamped on most tools. Therefore, even if the 200 μm bow limitation is important only at photolithographic steps, we will take it as a goal for all steps as a safety margin.

### 3.2. Stress and Wafer Breakage

Wafer bow is a signature of stress, in the deposited layer and in the wafer. But, to bow is not to break: silicon wafers can stand a large amount of stress without breaking (see pictures on Figure 7 below) whereas wafers without evident large stress may break catastrophically without solicitation. Silicon wafer are made of single crystal which yield by fracturing along crystallographic axis (cleaving) in presence of defects at the wafer edge. This is why wafer edges are rounded to avoid chipping, and except in presence of defects at its edges, a wafer will not break spontaneously, even if submitted to a consequent stress [21].

But the large bow observed in 3D technology is an aggravating factor (see below for details on technology). In the first fabrication run of 3D-DS detectors with 230-μm thick wafers, wafer yield was low because of the wafer breakage. The wafer breakage rate was not correlated with steps of higher bow, but was low at the beginning of the fabrication, and quite constant after the first Deep Reactive Ion Etching (DRIE). A rapid inspection of the wafer edge provided the cause of breakage: extremely damaged edges (see Figure 8a). Close inspection of the wafer edges at each steps of the DRIE mask definition brought the clue to this mystery (see Figure 8b–e): even without edge bevel removal, the resist does not cover fully the surface at the edge. Moreover, aluminum is not sputtered in a ring close to the edge. Therefore, some oxide can be etched during the DRIE mask definition (Al and SiO_2_ dry etch). As the wafer are thin and strongly warped during the electrodes DRIE, it seems that some free F* radicals can leak from the plasma below the chamber walls and reach the wafer edge to etch exposed silicon (silicon etch by F* radicals is a pure chemical reaction and does not require ion bombardment [22]): the edge is damaged and wafer can break in future manipulations and processes. This does not occur on standard, 525-μm thick wafers.

This problem is known in the literature and several solutions have been proposed: custom-made metal clamp on edge [23], deposition of a thick oxide hard mask on the edge, defined by an extra mask [24].

We found another solution without the need for tool modification, or extra mask, which is fully compatible with our process. Remember that silicon etch by fluorine radicals is chemical. Any layer not etched by these radicals would protect the exposed silicon on the wafer bevel. But this layer should be completely conformal to protect the wafer bevel, and should not need to be etched before the DRIE. We found that a thin (~10 nm) layer of alumina deposited by atomic layer deposition (ALD) just before DRIE completely fulfill these conditions. It is rapidly etched during the electrode DRIE, therefore only slightly delaying the electrode etch, and it is easily removed by a piranha bath normally used to strip the Al DRIE mask. After implementation of this solution, wafer yield increased from ~40% to 80% for 3D-DS process on 230 μm, to almost 100% for other technologies.

### 3.3. Application to Planar Process: Toy Model

Before dealing with the more complex case of 3D technology, let us first examine a standard planar p-n pixel process on 100 mm diameter wafers with two different thicknesses (230 μm and 285 μm). Wafer curvature (see Figure 9) mainly originates from two definite process steps: field oxide etching on the front and backside to define the front and backside contacts (+150 μm), and metallization (+90 μm). In order to model the bow variation during the process, we assumed that bow from silicon oxide and aluminum originates from thermal stress only (see Equation (5)). At sufficiently high temperature, silicon oxide is viscous enough to relieve all internal stress, due to volume mismatch between silicon and its oxide, leading to a pure thermal stress between growth and room temperatures [25]. For aluminum used in metallization, thermal stress usually dominates residual stress because of the large difference in coefficient of thermal expansion between aluminum and silicon (substrates in our sputtering tools are not cooled) [26].

The last point to be considered in this toy model is etching, which reduces the global bow to the fraction of the remaining layer (area coverage) [27]. Remember that stress in thermal silicon oxide will generate bow only when the oxide is etched (completely or partially) from one side of the wafer. Else, stress from films on both wafer sides balance and bow is null. Leaving apart some minor details of the process, we calculated bow from these assumptions for 230 μm thick wafer (see Table 2), and obtained a good agreement with the experimental value (see Figure 9). The only free parameter is the deposition temperature of aluminum. The obtained value of 100 °C agrees with the values obtained during simple deposition at IMB-CNM (80–100 °C) on similar wafers. Note that the deposition temperature depends critically on the wafer thicknesses: sputtering on 525 μm thick wafer lead to temperature increase of only 20 °C.

What we interpreted at first as bow increase of +90 μm due to metallization appears to be a roller-coaster ride, a combination and partial balance of front and backside metallization, and aluminum etching whose bow sums to −109.5 + 40.3 + 164.3 = 95.1 μm. Finally, the toy model can also be calculated for 285-μm thick wafers using the conversion factor 1.95/1.27 = 1.5 (see Figure 4), and again good agreement with experimental data is obtained.

## 4. Overview of Fabrication Technologies for 3D Detectors

### 4.1. Parker’s Proposal and First Productions at Stanford and SINTEF

3D radiation detectors were first proposed by Sherwood Parker in 1997 [1], and detailed account of the full process was given in a second paper [28]. These two papers contain all the basic and distinctive elements of the 3D technology:Deep reactive ion etching (DRIE) for the etching of high aspect ratio electrodes within the silicon volume.Polysilicon deposited by low pressure chemical vapor deposition (LPCVD) to ensure a conformal coating on electrode sidewalls and bottom, and a homogeneous dopant diffusion along the electrodes, due the high diffusion coefficient of dopants in polysilicon compared to silicon.Diffusion doping of p+ and n+ electrodes: these “old” processes are key to 3D technology since implantation is unable to provide a uniform doping in the high aspect ratio electrodes.

The process was completed at Stanford Nanofabrication Facility and starts with the fusion bonding of the active silicon n-type wafer to an oxidized support wafer to obtain a silicon-on-insulator (SOI) wafer (see Figure 10a). As this process was performed on n-type wafers, there were no need for p-spray, neither p-stop. Then, a 0.7 μm field oxide is grown and window for the n electrodes are defined by photolithography and etched in the field oxide (Figure 10b). The DRIE is performed following the Bosch process [29] with a 7 μm resist mask. Selectivity to resist is greater than 50 [1], diameter of electrodes is 11.5 μm, and electrodes are etched down the buried oxide (BOX). The electrode-doping process consists of two steps: first, polysilicon is deposited by LPCVD and afterwards phosphorus-doped with POCl_3_ source. The electrodes are then filled with polysilicon and polysilicon is etched away from both sides of the wafers. The same sequence is repeated for the p+ electrodes which are doped with a BBr_3_ source. Finally, aluminum-silicon alloy (Al/Si-1%) is sputtered and patterned on the front side.

The Stanford process is a single-side process with through-wafer electrodes: both electrode types are defined on the same side of the wafer, and both etched down to the buried oxide (BOX). It imposed the filling of the electrode holes, else the DRIE mask would not protect the first electrode holes from the second DRIE. Because of the original relatively large diameter of the electrode holes (11 μm), a large amount of polysilicon had to be deposited to fill the holes. The induced stress would cause wafer breakage in the first fabrication processes before the introduction of the handle wafer [30]. Process-induced stress was therefore already a concern. Wafer breakage was also observed during piranha cleaning, because of the thermal shock with the 125 °C hot and very reactive mixture, if the handle wafer was neglected [28]. Some actions were taken to reduce stress: recrystallization of polysilicon at elevated temperatures, and omission of electrode filling with polysilicon [28].

This process was then transferred to SINTEF MiNaLab in 2007, also on n-type wafers, in order “*to explore the possibility of small to medium size affordable 3D-detector production*” [31]. The process was therefore adapted to the specific tools of the SINTEF clean room. The main changes are:DRIE mask: Aluminum was substituted for thick resist, as neither resist nor silicon oxide offered sufficient selectivity for the long high aspect ratio etching [32];When the n-type electrodes were doped and filled, they were capped with a protection oxide of 300 nm. Note that polysilicon filling was still performed at Stanford Nanofabrication Facility because of limitations in the polysilicon process at SINTEF MiNaLab.A final passivation of 500 nm silicon oxide and 250 nm silicon nitride was deposited by plasma-enhanced chemical vapor deposition (PECVD) to guarantee protection from scratches, humidity, and contamination from outer world.

High stress was the main issue, aggravated by the use of automated tools and robots requisite for small- or medium-scale production. Wafer yield was low and accurate photolithographic alignment a challenge. They pointed to two origins of the high stress:Different oxide thickness between front and backsides of the wafers, leading to a bow around −130 μm [31]. Silicon nitride was deposited to avoid this asymmetry, and bow was reduced to +20 μm [33]. In both cases, wafer thickness was not directly specified, but if it was measured on the wafer mentioned in those papers (process wafer of 250 μm bonded on a 350 μm support wafer) these bow values imply really a large amount of stress, insuperable on thinner wafer, or after grinding of the support wafer;Polysilicon filling of the electrodes also induced a large stress, but this was mitigated by careful handling and planning [33].

### 4.2. Double Side Process (IMB-CNM, FBK)

3D Double Side (3D-DS) process was introduced by the Microelectronics Institute of Barcelona (IMB-CNM, CSIC) [34,35] and the Fundazione Bruno Kessler (FBK) [36]. (FBK and other foundries developed intermediate technologies; see [2] for a review). We will focus here on the 3D-DS technology developed at IMB-CNM for the Insertable B-Layer production (ATLAS) [3]. It was also installed in ATLAS Forward Proton (AFP) [4] and CMS-TOTEM Precision Proton Spectrometer (CT-PPS) [5].

Double side fabrication process avoids the support wafer and related bonding process, at the cost of a minimum processable wafer thickness (in practice, 200 μm for 100 mm). For IBL production, wafer thickness of 230 μm was agreed to be a best trade-off between the SNR and the mechanical robustness of the wafers for double-side processing [2]. Moreover, the backside is accessible for biasing. However, active edges cannot be implemented.

Process starts on 200–300 μm high resistivity p-type silicon wafers, with field oxidation and p-stop definition on the front side (Figure 11a). Then, aluminum is deposited on the backside to act as a DRIE mask. Once the aluminum and field oxide etched, p+ electrodes are defined by DRIE (Figure 11b). This is not a through-wafer etching: the p+ electrodes should be etched down to 20–30 μm from the other side of the wafer for optimum electrical performance. The etch depth is regularly controlled with a test wafer which is cleaved for microscope and Scanning Electron Microscope (SEM) inspection of the electrodes. After stripping of aluminum from the backside and appropriate cleaning, a 1-μm polysilicon layer is deposited by LPCVD and doped with BN (boron nitride) wafers on the backside (Figure 11c). In this technology, there is no need for filling the p+ electrodes since the other DRIE process will occur on the opposite side. The p+ polysilicon is protected from the subsequent n+ doping by a 200 nm grown oxide. Then, this process is repeated on the front side with POCl_3_ source for n+ doping, and polysilicon pads are defined on the front side to ensure a good electrical contact with metallization (Figure 11d,e). Finally, aluminum is sputtered and patterned, followed by a 400 nm-nitride/400 nm oxide passivation deposited by PECVD. For the IBL production, backside metal had to be patterned to define alignment marks, and a backside passivation was also added.

Pictures of finalized wafers can be seen on Figure 12a,b as well as zoom of FE-I4 pixel detectors (Figure 12c,d). Only the n+ electrodes can be observed from the front side.

The variation of wafer bow during the 3D-DS process is shown in Figure 13, with the bow variation of a planar pixel process for the sake of comparison. All curves are given for 230 μm thick wafers with 100 mm diameter. When compared to planar pixel abovementioned, complexity of bow variation for a 3D-DS process is striking (a real roller-coaster ride, this time!). This complexity reflects the complexity of the 3D-DS process in itself. The original 3D-DS technology (blue curve) was imported from the R&D phase on 285 μm wafers, without much consideration on curvature. Bow was therefore 1.95/1.27 = 1.5 less, due to difference in wafer thickness, and did not lead to major problems. That was not the case with the 230 μm: we discovered that a bow of ±300 μm is a practical limit of processability in most process tools. Actually, some wafers could not be processed during the N+ DRIE block when the absolute bow is maximum and close to 300 μm. Therefore, we decided to modify the process technology in order to reduce bow and improve manufacturability (red curve).

Several modifications were implemented. Let us consider first the metallization process. We described above a toy model for planar pixel detectors and its values for metallization should be usable here (see Figure 9 and Table 2). However, it does not reproduce experimental values for 3D-DS technology (dotted lines), maybe because of large curvature just before the metallization, leading to an incomplete thermalization of the wafer, and then higher temperature during sputtering. But still, it will help the discussion. In the original 3D-DS technology, a large bow >200 μm was expected before metal photolithography, value which exceeds the maximum value defined of 200 μm before. Therefore, we decided to deposit the backside metal just after front side metal and before the metal photolithography and etching. The bow was reduced (in absolute value) from >200 μm to 0–50 μm, reducing the overlay error during photolithography.

Second main stress contributor is the P+ doping block, for a total of −280 μm. It divides in two parts: polysilicon deposition and doping, which contributes to −130 μm, and the 200 nm oxidation (to protect p+ polysilicon from subsequent n+ doping), which contributes to −150 μm. Polysilicon deposition is known to lead to compressive stressed layers, with large variation depending on deposition conditions and annealing [37]. But we did not explore any variation in polysilicon deposition. Boron has a smaller covalent radius than silicon (0.97 Å versus 1.17 Å). Substitutional doping of boron into silicon will therefore result in a contraction of the lattice constant, leading to a strained layer in tensile stress (see Figure 2b above) [38]. We managed to reduce bow because of P+ doping from −130 μm to −15 μm by reducing the temperature of the BN doping, and therefore the doping of polysilicon.

What about the second part of the P+ doping block, the 200 nm protection oxide? Doping reduction mentioned above lead to a bow reduction from −150 μm to −100 μm, certainly because of the reason aforementioned, i.e., reduction in boron incorporation in grown oxide. However, bow estimation with values given for the planar pixel process indicates that a 200 nm oxide growth at 950 °C should induce a bow of 20 μm. How is it then that the measured bow ranges from 100 μm to 150 μm? The Stoney equation, which relates stress to wafer curvature, is only valid for planar technology, therefore it is inapplicable during the formation of the P+ and N+ columns. However, complex simulation would be needed to calculate the stress and bow induced by the tridimensional geometry; this is beyond the scope of this work (see however discussion on through-silicon via below).

Surprisingly, the bow during the N+ doping block is the same for the original and improved 3D-DS technologies. We cannot explain this behavior, but it illustrates the complexity of the stress in tridimensional structures. However, the improved technology fulfilled its role by reducing the wafer bow well below the 300 μm limit commented before.

### 4.3. 3D-SOI Technology at IMB-CNM

Forecast HL-LHC specifications for both ATLAS and CMS detectors draw toward thinner active regions, below the manageable thickness of 200 µm of the 3D-DS technology [6,7]. At IMB-CNM, we therefore decided to develop a single-side technology on SOI wafers (3D-SOI), like the Stanford/SINTEF technology, but armed with the experience of the 3D-DS technology [39,40]. The abovementioned reduction of BN doping thermal budget, for example, was implemented. Several differences with the Stanford/SINTEF technology should also be noted (see Figure 14 above):The p+ electrodes are not filled, but sealed, to minimize the amount of polysilicon deposited, and limit the wafer bow (see discussion below).The n+ electrodes are not filled: there is no further DRIE, so no risk of further etching; and the resist spins perfectly over the open narrow contacts, so that photolithography can be performed normally.The n+ electrodes are not etched down to the buried oxide, but 30 μm from it.The p+ electrodes are not accessed from the front side, but from the backside via a potassium hydroxide (KOH) etching of the support wafer and subsequent backside metallization.

This technology, like the 3D-SiSi technology described later, is a kind of hybrid: it is single-side from the point of view of manufacturing, and double-side from the point of view of the user.

This process has a main limitation with regard to a production: the final KOH etching of the support wafer is time-consuming and the resulting membranes can be quite fragile, lowering the yield. Moreover, the structured backside complicates further processing and photolithography, for example UBM deposition. This is why we switched to single-side technology on Si-Si wafers.

Photographs of finalized wafers and devices are shown on Figure 15.

On Figure 16, bow variation during fabrication process on SOI wafers is drawn. We use SOI with buried oxide of 1 μm, handle wafer of 300 μm, but three different active silicon thickness: 150 μm (blue curve), 100 μm (red), and 72 μm (green). It is clear that stress issues are less critical than on double-side technology on standard 230 μm wafers, mainly because of the thicker wafers used (372–450 μm), which reduce consequently the wafer curvature.

Note also that bow after KOH etching was not measured, therefore contribution of backside metal is not shown. Moreover, the initial thick backside oxide (~1.1 μm) is maintained through the whole process. This oxide is important for stress compensation and low initial bow value of the wafers. Its etching, added with backside metal deposition will surely induce a large bow (see 3D-SiSi process later).

Three steps contribute to the main part of the stress:Contrary to process on standard wafers, field oxidation contributes consequently to bow. A SOI wafer is not a perfectly strain-relaxed system, but a complex balance of compressive stress in the buried oxide and tensile stress in silicon [41]. Moreover, low to null initial bow in our wafers is achieved because of a ~1 μm backside oxide. Field oxidation at high temperature might relax part of the strain and break the initial balance.As in 3D-DS technology, a 200 nm oxide is thermally grown on the p+ doped polysilicon. It serves as an etch-stop for the undoped polysilicon etching (inter-polysilicon oxidation). It adds, depending on wafers, from +50 to +70 μm bow, which is slightly lower than measured on 3D-DS technology in the same conditions (−100 μm; the opposite sign comes from the opposite side of the contacts).Finally, the p+ pads are capped with a 500 nm thermal oxide to protect them from further etching and processes, and also to improve the voltage breakdown strength with nearby n+ metal pads. This step induces a bow of 35 to 90 μm depending on SOI wafers.

Solutions to reduce stress from the last two steps, and dependence on SOI wafer, will be discussed with the 3D-SiSi technology. But a difference in the bow patterns is noticeable: whereas bow in 3D-DS technology seems to compensate in a certain way (“roller-coaster ride”), in 3D-SiSi it seems to be cumulative (stepwise). This pattern may explain the initial easier manufacturability of the 3D-DS technology compared to the Stanford-SINTEF process.

### 4.4. 3D-SiSi Technology for High Luminosity LHC (HL-LHC)

Fabrication of 3D detectors on Si-Si bonded wafers (3D-SiSi) was first introduced by FBK [42]. These wafers offer the advantages of SOI wafers (mechanical resistance) without its drawbacks (KOH etching, anomalous stress behavior due to BOX). Direct wafer bonding (DWB) was first introduced for power applications (high voltage bipolar power transistors) to reduce wafer cost compared to thick high-resistivity epitaxial layer. Moreover, DWB wafers offer better thickness and resistivity control, and a relatively sharp transition region between the active p- active wafer and the p++ support wafer [43].

The fabrication developed at IMB-CNM is very similar to the SOI process (see Figure 17), except that access to the backside Ohmic contact is done through the handle p++ wafer. Like the SOI wafer process, depth of DRIE is key: p+ electrodes should reach the support wafer, while the n+ electrode should be etched 20–30 μm from the Si-Si boundary.

Bow variation during 3D-SiSi fabrication is shown in Figure 18 (blue curve). Bow variation for 3D-SOI of the same total thickness 350 μm and fabricated with the same technology is also shown for comparison (light blue curve). Similar patterns are observed, except for the bow due to field oxidation in 3D-SOI.

As in the previous technologies, the following steps contribute to most part of the measured bow: polysilicon deposition and doping (−80 μm), inter-polysilicon oxidation (+150 μm), capping oxidation (+170 μm), and backside metallization (+80 μm). The reduction of BN doping thermal budget, as discussed for 3D-DS technology, was implemented.

The 3D-SiSi is the technology chosen for the HL-LHC production. It is clear that the bow is too large with this total wafer thickness of 350 μm (150 + 200 μm), which is also the first choice for CMS, and would even be larger for the ATLAS production where the total wafer thickness will be of 270 μm (150 + 120 μm).

We have already noted that the Stoney equation, which relates stress to wafer curvature, is only valid for planar technology. However, clues can be found elsewhere: 3D detector technology holds many analogies with Through-Silicon-Via (TSV) technology [44]. Through-Silicon-Via are electrical interconnects etched with DRIE through the silicon wafers. They are basic pillars for 3D integration in advanced microelectronics. There are four main TSV technologies: via-first process, where the TSV is fabricated before the active devices; via-middle process, where the TSV are fabricated at the contact level; via-last process, where the TSV are fabricated at the end of the back-end-of-line process, and which divides into front side and backside via-last processes. Via-first processes use thermally grown oxide for insulation and doped polysilicon for filling. They are the closest to 3D detector technology, except that via diameter is normally large (>100 μm) to compensate for the relatively large resistance of the doped polysilicon. Via-middle and via-last processes use metals (W, Cu) for filling and allow for smaller via diameters. In all cases, strong stresses and warping were observed due to thermal mismatch between the silicon substrate and the materials composing the vias.

Numerous studies made use of finite element analysis to simulate stress around TSV and wafer warpage. Let us summarize some of the main results of interest for our application. Simulations and experiments clearly showed that thermal mismatch generates complex and non-uniform distribution of tensile and compressive stress along the column depth, and also along the wafer surface plane, with highly localized plastic deformation in a small region close to the via [45,46]. This justifies why the Stoney equation is not applicable, because it assumes a uniform stress along the whole layer surface. Moreover, stresses, and therefore warping, are higher in fully filled TSV than in annular (not filled) TSV [45]. This is our case in 3D-DS technology, and it might be one reason why high stresses developed in the original Stanford technology.

Che et al. [47] were able to simulate by finite element analysis warpage of an 8-in 750-μm thick silicon wafer during via-last TSV processing. Several of their results are worth noting for 3D technology. Wafer warpage increases with increasing TSV diameter and decreasing TSV pitch. The first is related to active silicon thickness and maximum DRIE aspect ratio achievable; the latter is defined by design and physics consideration. Wafer warpage is maximum when TSV depth is half the wafer thickness, and decreases when TSV depth/wafer thickness ratio decreases or increases. This dependence could explain the differences in bow observed in the various 3D technologies. They plotted wafer warpage as a function of TSV depth (see Figure 22 in reference [47]). But this representation does not allow to compare wafers with different thicknesses. Therefore, we decided to plot instead the bow variation multiplied by the wafer rigidity β_S_ (parametrized bow) versus the ratio of 3D column depth to wafer thickness (see Table 3 and Figure 19 below). Parametrized bow was referred to as stress-thickness product in planar technology when Stoney equation was valid.

On Figure 19, dotted lines are guides for the eyes. Simulated data by Che et al. [47] show a parabolic dependence of warpage on TSV depth and values of parametrized bow slightly higher, but the latter might be dependent on process. Our data seem to be closer to straight lines with relatively good fit, but the same trend is observable: bow is maximum when the TSV or 3D column depth is half the wafer thickness. Unfortunately, this is the choice made by ATLAS and CMS collaborations for the 3D production for the HL-LHC. While justified as a compromise between physics considerations and wafer strength, it is not favorable in terms of manufacturability. It makes bow reduction to be achieved only in terms of technological choices even more needed.

How could we reduce bow in 3D-SiSi technology? Pares et al. [48] found that, in their via-first process, stress was mainly due to thermal oxidation. They could reduce wafer bow replacing the thermal oxide by a high temperature deposited oxide. This deposition technique is not available at the IMB-CNM clean room. However, as stress in TSV and 3D columns is mainly due to thermal mismatch between silicon and the materials inside the columns, we decided to replace the thermal oxide by LPCVD silicon nitride (with low intrinsic stress). Silicon nitride CTE is close to silicon CTE (2.8 versus 2.5; see Table 1). Silicon nitride is also an appropriate etch-stop for the undoped polysilicon etching, which guarantees full process compatibility. Bow during inter-polysilicon oxidation, or better said inter-polysilicon dielectrics, was reduced from +150 μm to 0 μm (red curve in Figure 18)! Bow variation during capping oxidation was also reduced, without further process modification, from +170 μm to +80 μm. We believe this improvement is due to the close values of silicon and silicon nitride CTE, which limits stress rearrangement during further thermal processes.

We were surprised to find an increase in bow due to backside metallization, from +80 μm to +130 μm, when no modification was implemented at this stage of the fabrication. Maybe we should adjust the sputtering parameters for aluminum deposition to limit the intrinsic stress of the metal layers (see for example [12]).

## 5. Conclusions

3D detectors will enter soon a third phase in their development: after initial R&D phase and limited IBL production, larger production for HL-LHC will start soon. 3D technology was always scrutinized because of its increase complexity in fabrication. This article tackled this issue from the point of view of manufacturability, and especially wafer bow. We gave an overview of the various 3D detector technologies and improvements implemented to minimize the wafer curvature. We also singled out various parameters key to wafer bow: some dependent only on design (3D column density) and some related partly to technology (column diameter and depth, related respectively to active silicon thickness and to wafer total thickness). Finally, we were able to draw some analogies between 3D detector and Through-Silicon-Via (TSV) technologies.

3D detector technology was successfully extended at IMB-CNM to other devices, as U3DTHIN [49], neutron detectors [50], Vertical-JFET [51], and micro-dosimeters [52], on a variety of wafer types and thickness. But the highest curvatures were met in 3D detector technology, and the abovementioned technologies benefited the improvement implemented therein.

Bow is a good indicator of process issues and deviances. Bow is also obviously a signature of stress; stress can generate defects, such as dislocations, with direct implication on leakage current and therefore yield. This comes beyond the scope of this article, but further work should be dedicated to the study of local stresses and defect production, and their consequences on detector leakage current.

## Figures and Tables

**Figure 1 micromachines-11-01126-f001:**
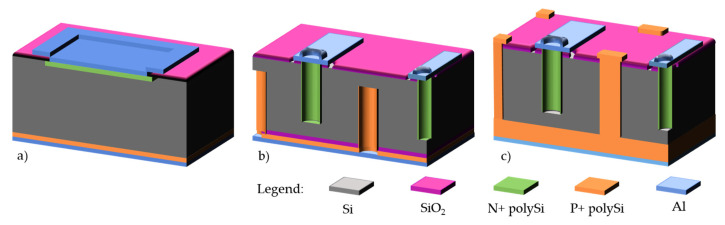
Comparison between different geometry of silicon radiation detectors: (**a**) planar, (**b**) 3D double-side, and (**c**) 3D-SiSi fabricated from Si-Si wafers.

**Figure 2 micromachines-11-01126-f002:**
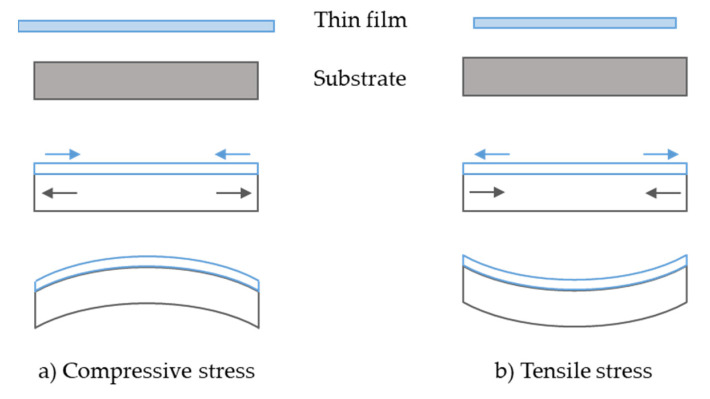
Imaginary sequence showing the building of (**a**) compressive and (**b**) tensile stress in thin films deposited on a thick substrate.

**Figure 3 micromachines-11-01126-f003:**
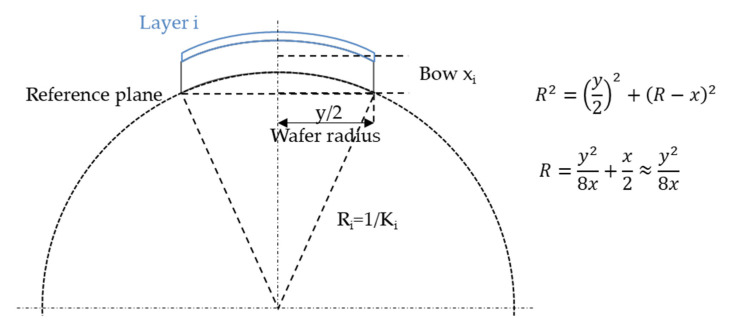
Relation between curvature radius and sagitta, or bow, of a wafer.

**Figure 4 micromachines-11-01126-f004:**
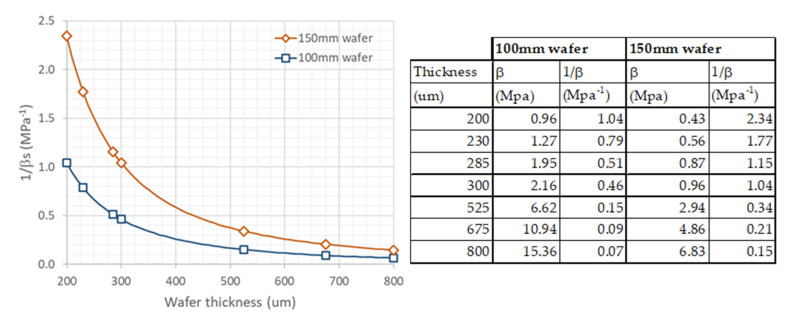
Variation of substrate parameter 1/*β_s_* with wafer thickness and values of *β_s_* for typical wafer thicknesses for 100 mm and 150 mm diameters.

**Figure 5 micromachines-11-01126-f005:**
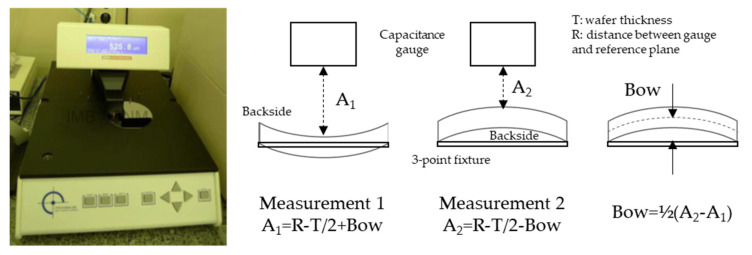
Bow measurement: picture of the Proforma300 in the Microelectronics Institute of Barcelona (IMB-CNM) clean room (**left**) and measurement method (**right**).

**Figure 6 micromachines-11-01126-f006:**
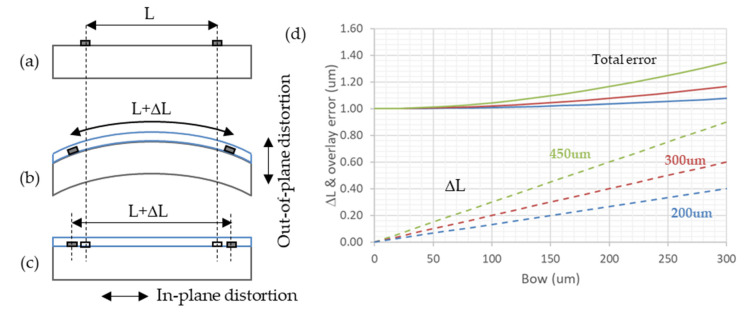
Effect of wafer bow on in-plane-distortion and overlay error (after [19]): (**a**) First alignment level on clamped wafer, with patterns separated by L; (**b**) deposition of a stressed layer, generating wafer distortion; (**c**) next alignment level on clamped wafer and in-plane-distortion; (**d**) in-plane-distortion and total overlay error (simple overlay error: 1 μm) as a function of wafer bow.

**Figure 7 micromachines-11-01126-f007:**
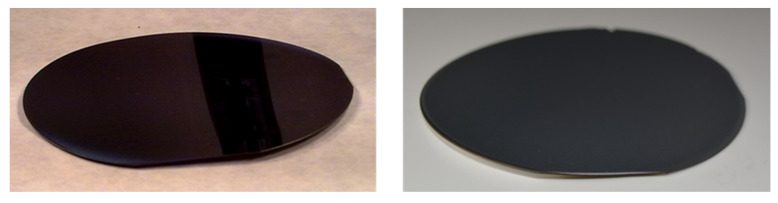
230 μm wafer after a not optimized boron diffusion doping (**left**) and 525 μm after a thick (~25 μm) aluminum sputtering (**right**) (courtesy of J. Montserrat, IMB-CNM, CSIC). Wafer deformation (warping) is clearly visible.

**Figure 8 micromachines-11-01126-f008:**
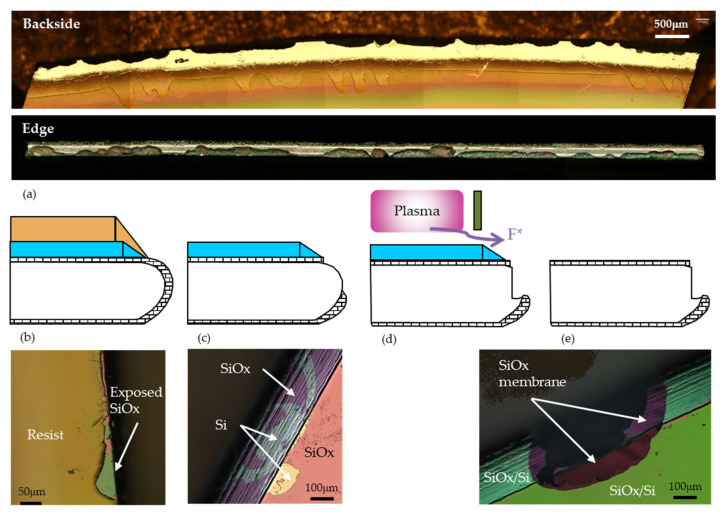
(**a**) Edge damage due to Deep Reactive Ion Etching (DRIE) observed from the etched side (backside) and the edge of the wafer; sequence of steps leading to the observed damage: (**b**) the resist does not cover the wafer surface to the edge, (**c**) silicon oxide is etched during definition of the DRIE mask, (**d**) exposed silicon is etched by F* radicals leaking form the plasma, and (**e**) final results: damaged edge.

**Figure 9 micromachines-11-01126-f009:**
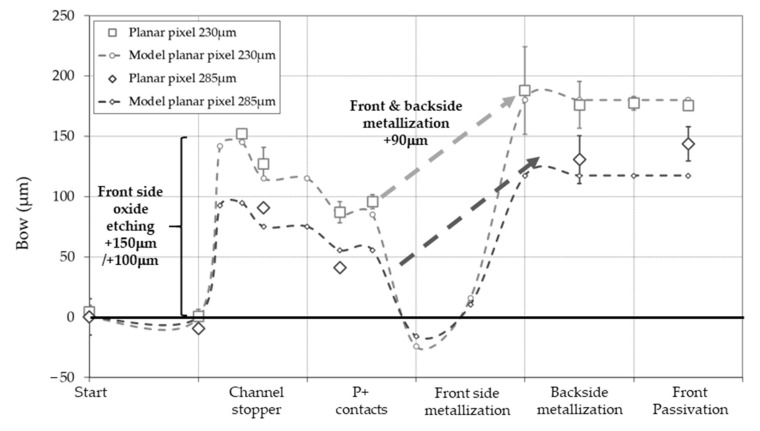
Bow variation during planar pixel process for two wafer thicknesses, 230 μm and 285 μm, and respective toy models (dotted lines). Main causes of curvature are indicated on the graph.

**Figure 10 micromachines-11-01126-f010:**
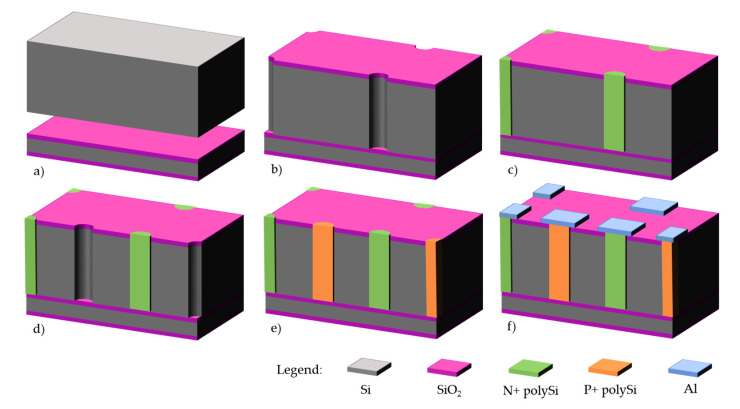
Stanford original process on n-type silicon wafers: (**a**) fusion bonding of oxidized support wafer with active wafer; (**b**) field oxidation, n+ window patterning and DRIE of n+ electrodes; (**c**) polysilicon deposition and n+ doping; followed by undoped polysilicon filling; (**d**) pattern and etching of p+ window, and DRIE of p+ electrodes; (**e**) polysilicon deposition and p+ doping; followed by undoped polysilicon filling; (**f**) metal deposition and patterning. Vertical and horizontal scales are different, and support wafer is not to scale.

**Figure 11 micromachines-11-01126-f011:**
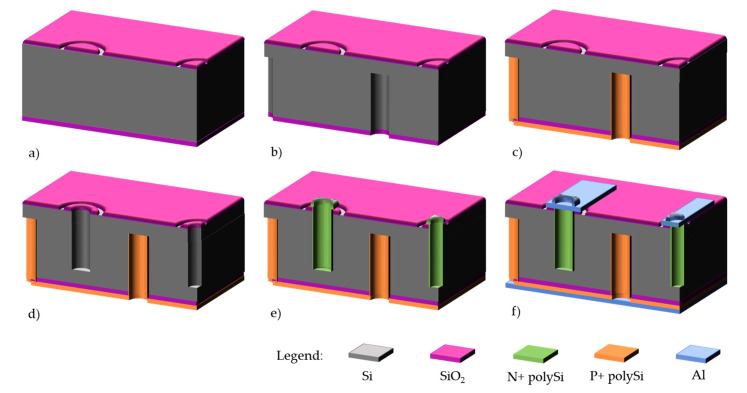
Double-side process at IMB-CNM on p-type wafers: (**a**) field oxidation and p-stop definition; (**b**) p+ window patterning and DRIE of p+ electrodes on backside; (**c**) polysilicon deposition and p+ doping; (**d**) n+ window patterning and DRIE of n+ electrodes on front side; (**e**) polysilicon deposition and n+ doping; (**f**) front and backside metallization and patterning. Passivation and temporary metal are not represented for clarity.

**Figure 12 micromachines-11-01126-f012:**
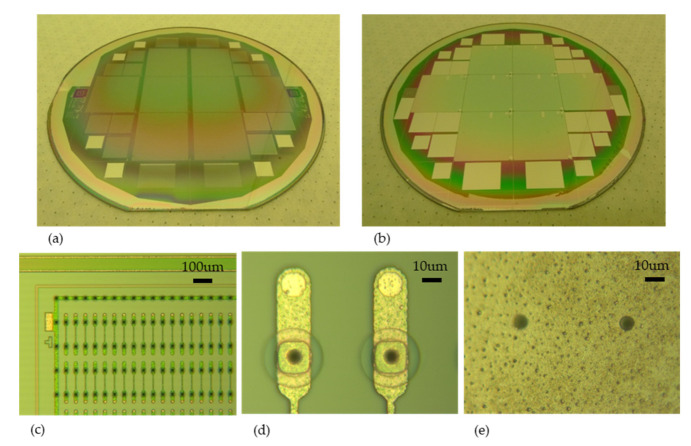
Front side (**a**) and backside (**b**) photographs of IBL wafers. View of FE-I4 devices, front side (**c**,**d**) and backside (**e**) from ATLAS Forward Proton(AFP) production.

**Figure 13 micromachines-11-01126-f013:**
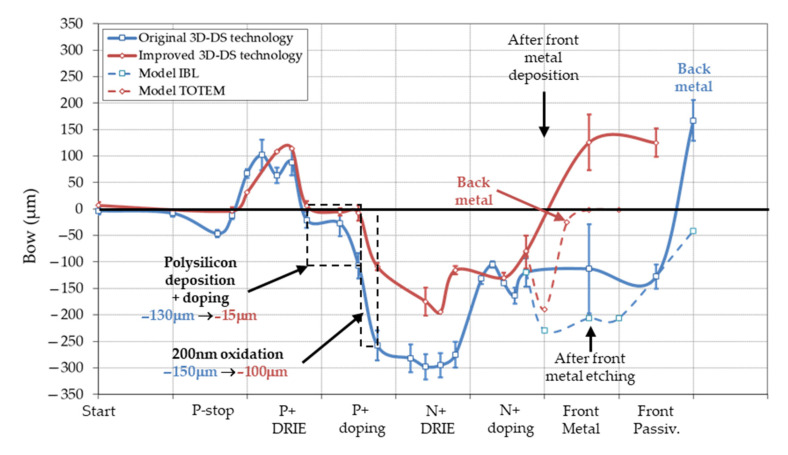
Bow variation during two 3D-DS processes (wafer thickness: 230 μm) and planar models for the metallization. Main causes of curvature are indicated on the graph.

**Figure 14 micromachines-11-01126-f014:**
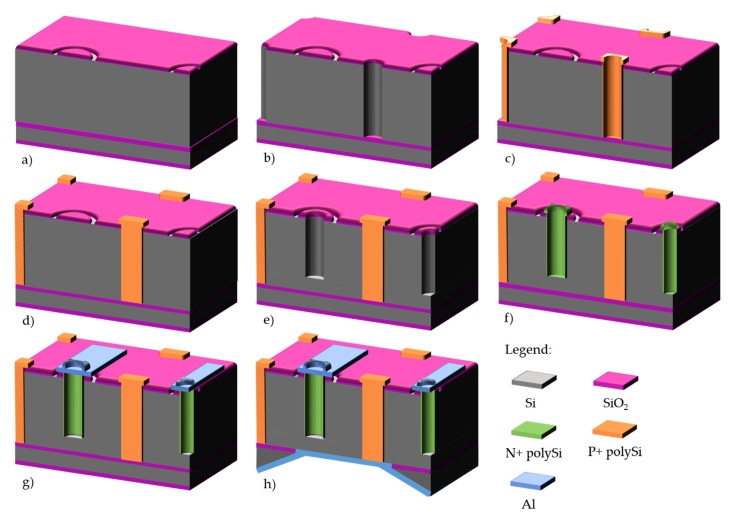
IMB-CNM 3D-SOI technology on SOI p-type wafers: (**a**) field oxidation and p-stop definition; (**b**) p+ window patterning and DRIE of p+ electrodes; (**c**) polysilicon deposition and p+ doping; (**d**) sealing of p+ electrodes with undoped polysilicon; (**e**) pattern and etching of n+ window, and DRIE of n+ electrodes; (**f**) polysilicon deposition and n+ doping; (**g**) metal deposition and patterning on the front side; (**h**) support wafer etching with KOH from the backside and metallization. Fabrication of SOI wafers is done externally and therefore not included in the process.

**Figure 15 micromachines-11-01126-f015:**
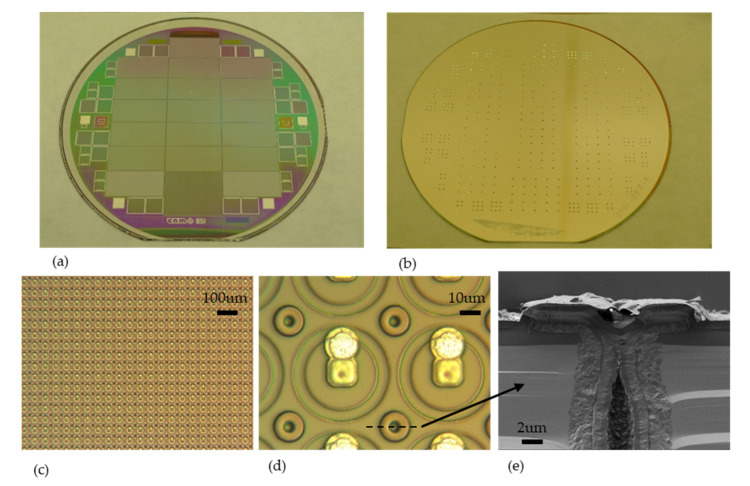
Front side (**a**) and backside (**b**) photographs of single-side SOI wafers. Pads from p+ electrodes are visible on the front side. View of RD53A 50 × 50 μm^2^ devices (**c**,**d**) and Scanning Electron Microscope (SEM) cross-section of p+ pad (**e**).

**Figure 16 micromachines-11-01126-f016:**
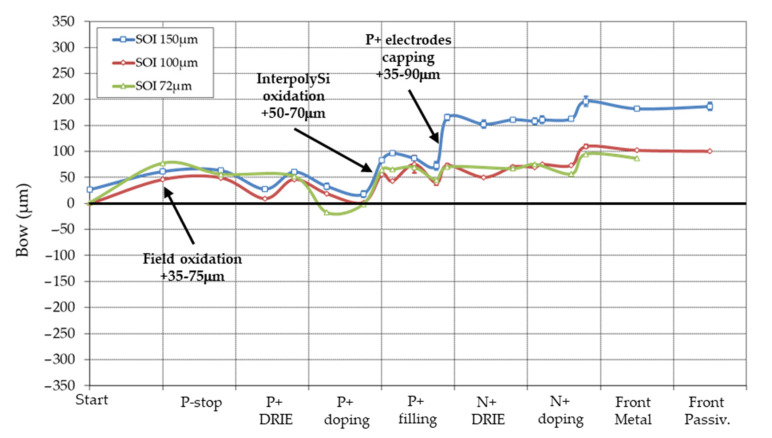
Bow variation during process of SOI wafers of various silicon device thicknesses. Handle wafer: 300 μm, BOX: 1 μm. Main stress contributors are indicated on the graph.

**Figure 17 micromachines-11-01126-f017:**
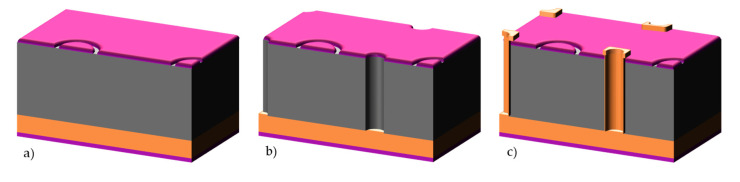
IMB-CNM 3D-SiSi technology on Si-Si p-type wafers: (**a**) field oxidation and p-stop definition; (**b**) p+ window patterning and DRIE of p+ electrodes; (**c**) polysilicon deposition and p+ doping; (**d**) sealing of p+ electrodes with undoped polysilicon; (**e**) pattern and etching of n+ window, and DRIE of n+ electrodes; (**f**) polysilicon deposition and n+ doping; (**g**) front and backside metal deposition and patterning on the front side; (**h**) Deposition and patterning of temporary metal for electrical characterization of pixel detectors. Fabrication of Si-Si wafers is done externally and therefore not included in the process.

**Figure 18 micromachines-11-01126-f018:**
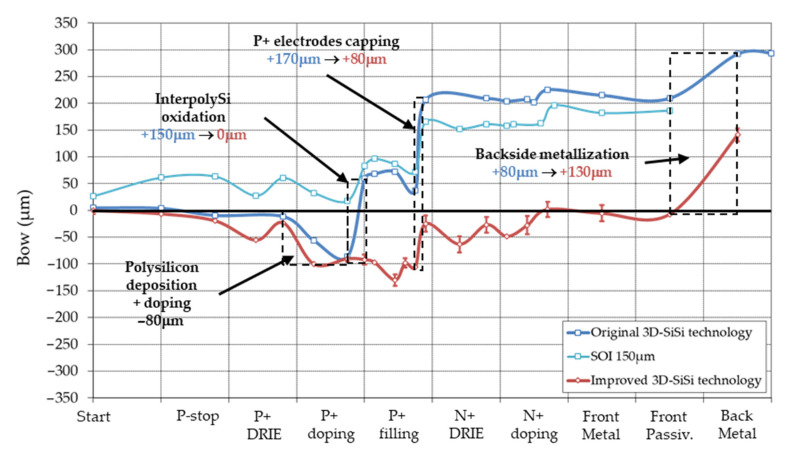
Bow variation during process of Si-Si wafers with 150 μm active silicon (350 μm total thickness) for original (blue) and improved technologies (red). Bow variation for a 150/1/300 μm SOI wafer is shown for comparison (light blue). Main stress contributors for SiSi wafer are indicated on the graph.

**Figure 19 micromachines-11-01126-f019:**
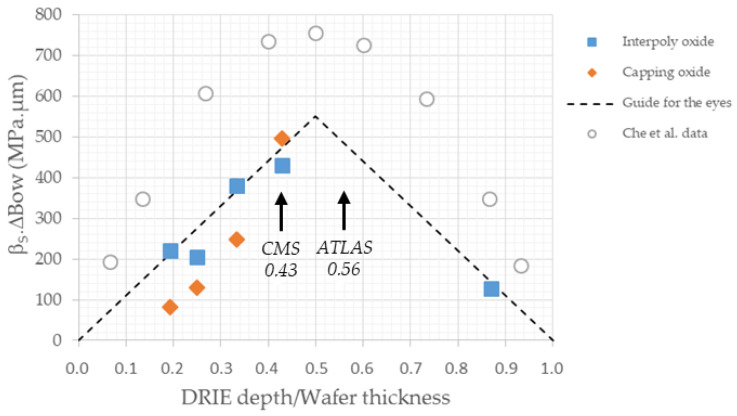
Parametrized bow versus DRIE depth to wafer thickness ratio of critical steps in 3D detector technology. Dotted lines are guide for the eyes. Data from [47] are reproduced for comparison. Values of DRIE depth to wafer thickness ratio for ATLAS and CMS HL-LHC production are also indicated.

**Table 1 micromachines-11-01126-t001:** Coefficient of thermal expansion and biaxial elastic module G of various materials used in 3D technology.

Materials	CET	B	Ref
	10^−6^K^−1^	GPa	
Silicon	2.5	180	[Hopcroft10]
Thermal SiO_2_	0.5	164	[Sinha78]
LPCVD Si_3_N_4_	2.8	3500	[Sinha78]
Aluminium	23.2	84	[Jamting97]

**Table 2 micromachines-11-01126-t002:** Toy model for bow during the fabrication of a planar pixel detector (wafer thickness: 230 μm).

Step	Layer					Covered Area		Stress (Mpa)		∆Bow
	Material	Thickn.	Temp.	CTE	B	Front	Back	Front	Back	(µm)
		(nm)	(°C)	(10^−6^K^−1^)	(GPa)					
Process start	Silicon			2.5				0.0	0.0	0.0
Field oxidation	Silicon oxide	800	1100	0.5	164	100%	100%	354.2	354.2	0.0
Contact definition	Silicon oxide	800	1100	0.5	164	64%	0%	354.2	0..0	141.9
Front metallization	Aluminium	1000	100	23.2	84	100%	0%	−139.1	0.0	−109.5
Front metal etching	Aluminium	1000	100	23.2	84	37%	0%	−139.1	0.0	40.3
Backside metallization	Aluminium	1000	100	23.2	84	0%	100%	0.0	−139.1	164.3

**Table 3 micromachines-11-01126-t003:** Estimation of parametrized bow in critical steps of 3D technologies.

					InterPolySi Oxide		Capping Oxide	
Technology	Wafer Total Thickness (µm)	3D P+ Column Depth (µm)	Ratio	Βs (MPa)	∆Bow (µm)	Βs. ∆Bow (Mpa. µm)	∆Bow (µm)	Βs. ∆Bow (Mpa. µm)
3D-DS improved	230	200	0.87	1.27	101.1	128.4		
3D-SOI 150mm	450	150	0.33	4.86	78.4	380.9	88.1	428.3
3D-SOI 100mm	400	100	0.25	3.84	53.1	203.9	33.7	129.4
3D-SOI 72mm	372	72	0.19	3.32	66.2	219.7	25.0	83.0
3D-SOI original	350	150	0.43	2.94	146.0	429.2	168.8	496.3

*Bow given in absolute value*.

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
