# Peer review of "Manufacturability and Stress Issues in 3D Silicon Detector Technology at IMB-CNM"

_micromachines, 2020, doi:10.3390/mi11121126_

Round 1
Reviewer 1 Report
The article is well written, coherent and well-structured. It requires only a slight review of English and some minor revision.
fig.4. increase the size of plot and table. Make the plot more readable for gray level visualization, possibly using darker colors
pag.11: 342. cancel 'a' before 'polysilicons'
pag.15: 451. 'induce'
pag.16:459. adjust the reference
caption fig 19: 'ratio'
Author Response
Dear reviewer,
Thank you very much for your positive review.
Your revisions have addressed in the updated manuscript.
Best regards
David Quirion
Reviewer 2 Report
The manuscript "Manufacturability and Stress Issues in 3D Silicon Detector Technology at IMB-CNM" was an excellent read and written in an easy to read style. I appreciate the authors sharing of their overview of 3D detector fabrication technology at the IMB-CNM. The value of these radiation hardened low power consumption 3D detectors is becoming very clear. The material stresses associated with building these 3D electronic structures of a variety of components is clearly an issue with defective devices. The authors have clearly explained and built a model for how these stresses come into play with wafer bow and other thermal stresses. It is great to see that on-line bow mentoring integration into the fabrication process. The authors explanation of the process and solutions will be invaluable for others trying to build 3D silicon detectors. I recommend this manuscript for publication with a very minor edit to the text starting at line 213.
The introductory sentence is written in an odd fashion with a double negative... It likely should read.
231 "We found another solution without the need for tool modification, or an extra mask, which is fully compatible with our process"
Author Response
Dear reviewer,
Thank you very much for your positive review.
Test starting at line 213 was changed in the updated manuscript.
Best regards
David Quirion